# Sugary interfaces mitigate contact damage where stiff meets soft

Hee Young Yoo[1,*], Mihaela Iordachescu[1,*], Jun Huang[2], Elise Hennebert[3,4], Sangsik Kim[5], Sangchul Rho[6], Mathias Foo[7], Patrick Flammang[3], Hongbo Zeng[2], Daehee Hwang[6,8], J. Herbert Waite[9] & Dong Soo Hwang[1,5]

The byssal threads of the fan shell *Atrina pectinata* are non-living functional materials intimately associated with living tissue, which provide an intriguing paradigm of bionic interface for robust load-bearing device. An interfacial load-bearing protein (*A. pectinata* foot protein-1, apfp-1) with L-3,4-dihydroxyphenylalanine (DOPA)-containing and mannose-binding domains has been characterized from *Atrina*'s foot. apfp-1 was localized at the interface between stiff byssus and the soft tissue by immunochemical staining and confocal Raman imaging, implying that apfp-1 is an interfacial linker between the byssus and soft tissue, that is, the DOPA-containing domain interacts with itself and other byssal proteins via $Fe^{3+}$–DOPA complexes, and the mannose-binding domain interacts with the soft tissue and cell membranes. Both DOPA- and sugar-mediated bindings are reversible and robust under wet conditions. This work shows the combination of DOPA and sugar chemistry at asymmetric interfaces is unprecedented and highly relevant to bionic interface design for tissue engineering and bionic devices.

[1] Division of Integrative Biosciences and Biotechnology, Pohang University of Science and Technology, Pohang 37673, Republic of Korea. [2] Department of Chemical and Materials Engineering, University of Alberta, Edmonton, Alberta, Canada T6G 2V4. [3] Laboratory of Biology of Marine Organisms and Biomimetics, Research Institute for Biosciences, University of Mons, 7000 Mons, Belgium. [4] Laboratory of Cell Biology, Research Institute for Biosciences, University of Mons, 7000 Mons, Belgium. [5] School of Environmental Science and Engineering, Pohang University of Science and Technology, Pohang 37673, Republic of Korea. [6] Center for Plant Aging Research, Institute for Basic Science (IBS), Daegu 42988, Republic of Korea. [7] School of Engineering, University of Warwick, Coventry CV4 7AL, UK. [8] Department of New Biology, DGIST, Daegu 42988, Republic of Korea. [9] Department of Molecular, Cellular, and Developmental Biology, University of California, Santa Barbara, California 93106, USA. * These authors contributed equally to this work. Correspondence and requests for materials should be addressed to D.S.H. (email: dshwang@postech.ac.kr).

Achieving robust attachment between living tissues and inert materials faces the challenge of mitigating contact damage due to mismatches in critical properties such as stiffness, hardness, strength and Poisson's ratio[1,2]. Contact damage mitigation is especially critical for durable loadbearing prosthetic or implant devices. Prosthetic devices for amputees, for example, are of tremendous benefit to mankind, but contact damage such as inflammation, rashes, blistering and numbness can limit their usefulness to the fullest potential[3]. There is growing belief that the best engineered interfaces are those inspired by nature (bionic), such as between bone, and muscle[4,5], nail and skin[6], and more recently, squid beak and muscle[7]. These natural interfaces are robust, dynamic and durable—all desirable properties for bionic devices—but on their transferability to engineer bionic devices have yet to be fully understood[8]. In particular, the molecular level characterization of the interface remains elusive.

From a mechanical perspective, there appear to be two strategies in biology for mitigating contact damage between mismatched loadbearing materials: (A) using a strong interfacial adhesion over an increased surface area of contact (thereby reducing the load per unit area), or (B) creating an interface that relies on molecular gradients. The former, suitable for bulky materials, is the strategy employed in the tendon–muscle junction[4,5].

In the latter, the stiffer material is gradually ramped down to the more compliant one by decreasing the ratio of hard to soft blocks in naturally occurring block copolymers, such as in *Mytilus* byssus and polychaete jaws[1,2,7,9]. The butt-joint model[10] predicts that, as the stiffness of the two contacting materials begins to coincide, the radial stress goes to zero. Contributions of the Poisson ratio, another factor in this model but difficult to measure accurately in small samples, are usually ignored.

Our work on the fan shell mussel, *Atrina pectinata* L., sheds some light on the molecular interface between living tissue and load-bearing non-living material. *Atrina* uses a byssus to anchor itself to available hard objects in the benthic sediment[11,12]. About a proximal third of each byssal thread is rooted in the soft tissue within the shell (Fig. 1a), while the rest is exposed outside the shell for attachment to pebbles or detritus. The dimensions of the exposed thread are about 25 cm in length and 25 µm in diameter[11]. Interestingly, the rooted thread portion originates in the byssal adductor muscle and proceeds from there along the byssal groove till it emerges from the living tissue and the shell (Fig. 1a,b). Water flow over the exposed shell results in lift and drag forces that tug on the attached threads. The threads transfer and dissipate the loads to the interconnecting tissue without incurring any damage[12–15]. In this work we report insights at a molecular level about the interface between tissue and embedded byssal threads and how this affects tenacity, toughness, and robustness of a bionic holdfast. The characterized *Atrina*'s foot protein has L-3,4-dihydroxyphenylalanine (DOPA) containing domain and sugar-binding domain that is localized at the interface between tissue and embedded byssal thread. Our results show that the combination of this DOPA-containing and sugar-binding domain plays a significant role in achieving robust load-bearing device and this unprecedented finding has high relevance to bionic interface design for tissue engineering and bionic devices.

## Results

**Mechanical mismatch between living and non-living material.** Previous studies on byssal threads in the genus *Mytilus* invoked molecular gradients along the axis of each thread to produce a stiffness gradient to moderate the stiffness mismatch between the

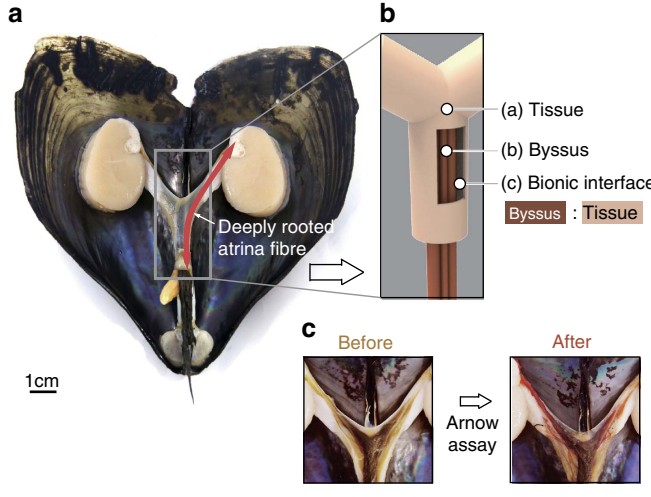

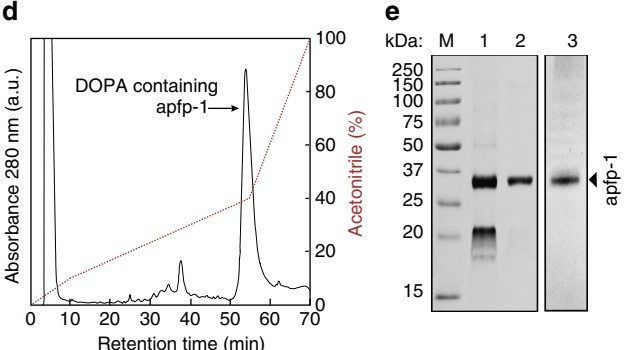

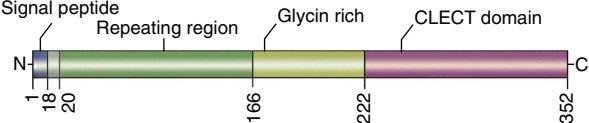

**Figure 1 | Characterization of *Atrina pectinata* foot protein-1 (apfp-1).** (**a**) Opened fan shell showing the full body after removal of the buccal mass to highlight the deeply rooted byssus through the body. (**b**) Schematic illustration of the dissected byssus wrapping tissue (**c**) Byssus in **b** after staining for DOPA-containing proteins with a catechol-specific reagent. (**d**) C8 HPLC chromatographic separation of apfp-1. (**e**) Gel electrophoresis purification of apfp-1. M: Molecular mass markers, 1: fractions corresponding to GPC major peak; 2: HPLC major peak obtained after injection of the GPC main peak, 3: Catechol-specific staining of apfp-1. (**f**) Schematic representation of apfp-1. Functional domains are drawn to scale at the location within the full protein sequence.

threads and where these threads meet in the stem before entering the living tissue[13]. Specifically, two hybrid collagens in each thread, preCOL-D (collagen + silk domains) and preCOL-P (collagen + elastin domains), are self-assembled in such a way that the stiffer preCOL-D predominates distally, whereas the more compliant preCOL-P prevails approaching the living tissue[13–15]. There is no evidence for molecular gradients in *Atrina* threads; each hydrated thread has a uniform stiffness that is ∼ 100 times stiffer than the surrounding tissue (Supplementary Table 1; Supplementary Discussion)[16]. The nano-indentation measurements have also confirmed the absence of this molecular gradient with its cuticle and core having indistinguishable mechanical property (Supplementary Table 2). As such, this

stiffness mismatch almost certainly foreshadows contact damage where thread meets tissue, unless geometry and the large interfacial contact area are designed to dissipate energy efficiently. The *Atrina* byssus meets living tissue in a joint that is very different in geometry from the molecular and mechanical gradients[13–15,17] in *Mytilus* threads. *Atrina* threads exploit the high surface area associated with embedding nearly 10 cm of each thread in the byssal groove. To this, *Atrina* adds numerous sacrificial, strong and reversible lectin-type interactions across the thread–tissue interface to dissipate energy during load transfer.

**DOPA as a key adhesive component of the byssus**. The chemistry of *Mytilus* byssus has been well studied[18]. A key adhesive signature of the byssus is DOPA, a catecholic amino acid that is post translationally modified from tyrosine[18–20], that is also present in *Atrina* byssus (Supplementary Table 3; Supplementary Discussion). DOPA-containing proteins in *Mytilus* byssus play a key role in both underwater adhesion and load bearing of the byssus[20,21]. Specifically, DOPA in the mussel adhesive proteins forms adhesive bonds with multivalent ions, metal oxides and organic functional groups, which lead to the byssus having robust, stiff and extensible character in underwater adhesion; a vital trait for load bearing by *Mytilus* byssus. $Fe^{3+}$ ions, which are $10^6$ fold more enriched in the *Mytilus* byssus than in the ambient seawater, form strong and reversible complexes with DOPA for enhancing the mechanical properties of the *Mytilus* byssus[22–24].

Similar to its *Mytilus* counterpart, the *Atrina* byssus is also predominantly protein (97 wt%, Supplementary Table 4) with abundant metal ions that potentially interacts with DOPA. Additionally, from inductively coupled plasma mass spectrometry (ICP-MS) analyses, we detected a substantial metal ion content, primarily $Ca^{2+}$ and $Fe^{3+}$, at levels that are $10^3$–$10^6$-fold more enriched in the *Atrina* byssus than in seawater (Supplementary Table 5; Supplementary Discussion). The high metal ion content draws attention to their possible role of mediating protein interactions in the byssus. Given the known interaction between DOPA and $Fe^{3+}$, we proceeded to look for DOPA-containing proteins in the *Atrina* byssus. The byssus was dissected and, following Arnow staining, DOPA was shown to occur throughout the *Atrina* byssus (Fig. 1c).

**Interfacial protein *Atrina pectinata* foot protein-1**. Having determined the presence and distribution of DOPA in *Atrina*, we moved on to the purification of the DOPA-containing proteins. The *Atrina* foot proteins were extracted from the foot by using 5% acetic acid, and thereafter, proteins from the *Atrina* foot extract were separated by gel permeation chromatography (GPC) and then subjected to C8 reverse phase high-performance liquid chromatography (HPLC). A major peak was eluted at ∼48% (v/v) acetonitrile (Fig. 1d). The major peak was run on SDS–PAGE (polyacrylamide gel electrophoresis) and a dense single band of apparent mass ∼36 kDa was observed in the gel (Fig. 1e). Acid urea PAGE (AU-PAGE) of the major peak showed a nitro-blue-tetrazolium-positive band, indicating the presence of DOPA in the purified proteins (Fig. 1e). The major peak was also subjected to matrix-assisted laser desorption and ionization with time-of-flight (MALDI-TOF) mass spectrometry yielding a mass (*m/z*) of 38,705.4 Da (Supplementary Fig. 1) that is consistent with the apparent mass from SDS–PAGE.

The purified DOPA-containing protein (∼38 kDa) was transferred to a polyvinylidene difluoride membrane (Immobilon P) and subjected to ESI-MS/MS (electrospray ionization-mass spectrometry) peptide sequencing after *in situ* trypsin digestion. Three peptides were identified following the collision induced

dissiciation by ESI-MS/MS: ODYKOVPK (O denotes hydroxylproline) (Supplementary Fig. 2), YELKPGVW, and GLNDVLFPK. In order to obtain a partial cDNA-deduced sequence of the *Atrina* foot protein, 3′ RACE was performed using degenerate oligonucleotides designed from the known amino-acid sequence of the tryptic peptide PDYKPVPK and the Universal Primer Mix. After cloning and sequencing the partial cDNA, a specific primer was designed to amplify the 5′ end of the gene and to retrieve the full-length cDNA. The obtained sequence of the cDNA has a length of 1,071 bp, which translates into a protein of 360 amino acids in length (Supplementary Fig. 3). The predicted molecular mass and theoretical pI of the cDNA-deduced amino-acid sequence without the signal peptide were 38,485.8 Da and 9.47, respectively. The cDNA-deduced amino-acid composition without the signal peptide matched with the amino-acid composition of the protein isolated from the major HPLC peak (∼38 kDa, Supplementary Table 3). In addition, the 38 kDa protein from the major peak was subjected to Edman N-terminal sequencing and revealed ASXVPPVD (where X denotes an unknown amino-acid residue) as the starting sequence, which matched that of the cDNA-deduced sequence after the signal peptide. The complete cDNA-deduced protein was named *A. pectinata* foot protein-1 (apfp-1). The mass difference between apfp-1 by MALDI-TOF mass spectrometry and the cDNA-deduced apfp-1 sequence without the signal peptide was around ∼220 Da, suggesting apfp-1 has around 13–14 DOPA residues if all the post-translational modification corresponds to DOPA. This, however, is not likely, given that at least two proline residues are also hydroxylated. On the other hand, only trace amounts of hydroxyproline were detected by amino-acid analysis suggesting that Hyp in apfp-1 may be limited to the peptide ODYKOVPK. DOPA content calculated from MALDI-TOF roughly resembled the amino-acid analysis. DOPA content of 38 kDa bands (apfp-1) was 5.2 ± 0.5 mol% based on amino-acid analysis, which suggest 16–19 DOPA residues in native apfp-1 (Supplementary Table 3).

apfp-1 has four distinct regions (Fig. 1f). The first region beginning with the N terminus, is a signal peptide sequence 18 residues long. A domain with degenerate tandem repeats follows including the octapeptide VVPDYKPP, hexapeptide VPKYKS/P along with the heptapeptide PVDYKPP, and a tetrapeptide PVYK that is repeated six times. Similar consensus repeats are also found in other mussel foot protein-1 (mfp-1) analogues[18,20] (Supplementary Table 6). After the repeating region, there is a 55 amino-acid-long glycine-rich region. Within this region, there are nine tyrosine residues, mostly flanked by glycine. This arrangement poses less steric crowding for the DOPA residues and should result in better solvent accessibility, Fe-coordination and catalytic conversion by catechol oxidase of Tyr to DOPA, and later DOPA to Dopaquinone, which is similarly observed in proteins from other organisms including *Aulacomya ater*, *Trichomya hirsute*, *Fasciola hepatica* and *Phragmatopoma californica*[25–27]. The C-terminal domain of the protein is predicted to be a C-type lectin domain, which has a carbohydrate-binding motif[28]. The presence of the C-type lectin domain accounts for the substantial amount of $Ca^{2+}$ detected in our ICP-MS analysis as $Ca^{2+}$ is known to be the metal cofactor in the binding of lectin and sugar[28].

DOPA–$Fe^{3+}$ complexes have a unusual high-stability constant[22,24,29]. Coordination-based crosslinks have been proposed to endow certain biological structures with a number of desirable material properties, including triggered self-assembly, increased toughness, self-repair, adhesion, high hardness in the absence of mineralization and mechanical tunability[18,21–24,29]. The formation of $Fe^{3+}$–DOPA complexes was detected by resonance Raman spectroscopy[23,29] of purified apfp-1 protein

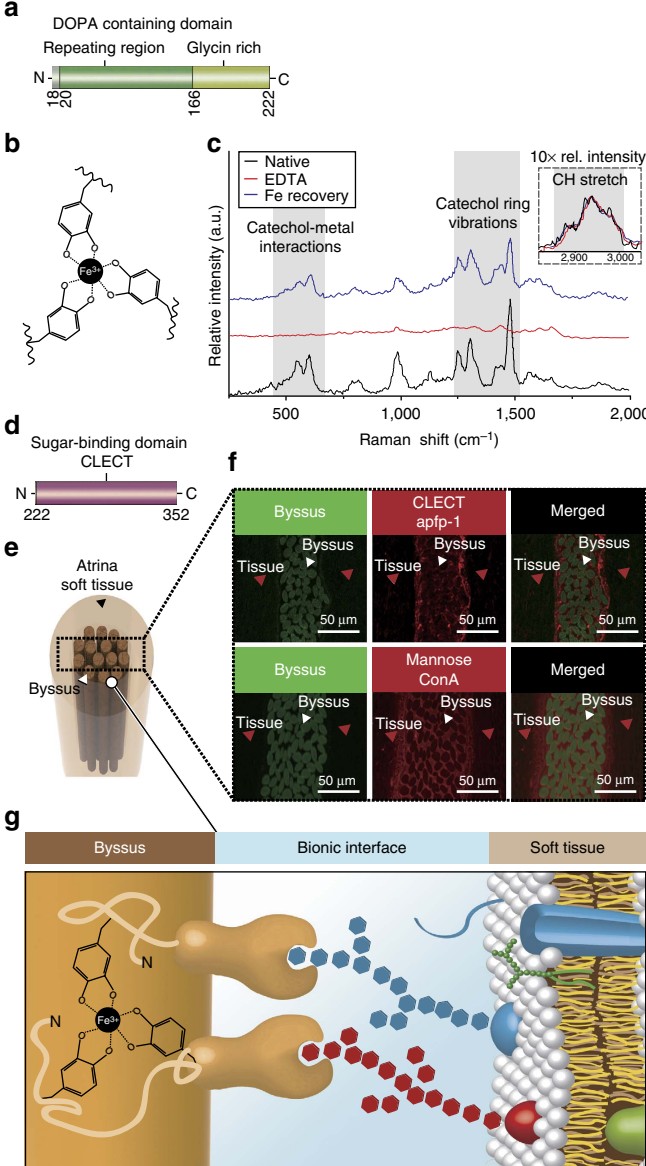

**Figure 2 | Role of apfp-1 in *Atrina pectinata*.** (**a**) DOPA-containing domain in apfp-1. (**b**) Tris-DOPA–$Fe^{3+}$ complex. (**c**) Resonance Raman spectra from apfp-1 (black), after EDTA treatment (red) and after re-exposure to Fe in apfp-1 (blue). The nearly complete loss of resonance peaks after EDTA treatment is reversed to a nearly complete restoration by means of incubation. in 1 mM $FeCl_3$ (pH 3.2). The three spectra were normalized to the area under the aliphatic CH peak (2,850 to 3,010 $cm^{-1}$). The inset represents the relative intensity of the non-resonance peak for aliphatic CH stretching from apfp-1 magnified $\times$ 10. According to the assignments, the most prominent peaks can be attributed to the interaction of metal with the catecholic oxygens and to the vibrations of the carbon bonds in the catechol ring, respectively. (**d**) Sugar-binding domain in apfp-1. (**e**) Cross-section through the tissue surrounding the *Atrina* byssus. (**f**) Confocal imaging of *Atrina* byssus and the surrounding tissue labelled with anti-apfp-1 antibody to detect apfp-1, and with ConA to detect mannose. Green box represents the auto-fluorescence of the byssus which is the location that shows where the byssus fibre is. Red box represents the apfp-1 antibody to detect apfp-1, and with ConA to detect mannose. (**g**) The byssus is coated with apfp-1. Its DOPA–$Fe^{3+}$ binding property gives stiff and extensible properties to the cuticle region, like fp-1 in mussel byssus. The apfp-1 lectin domain strongly binds to mannose in the cell membrane that surrounds the non-living byssus and acts as the bionic interface between non-living tissue and the cell.

(Fig. 2b,c). Three diagnostic regions of Raman spectra were observed: catechol–$Fe^{3+}$ interaction (550–680 $cm^{-1}$) and catechol ring vibrations (1,240–1,370 $cm^{-1}$) are resonance dependent where they are associated with the transfer of an electron from catechol to $Fe^{3+}$, and CH stretching (2,850–3,010 $cm^{-1}$), which is not resonance dependent. The Raman spectra acquired from protein purified in the presence of EDTA, which is used to scavenge $Fe^{3+}$ from complexation with catechol showed the loss of $Fe^{3+}$ associated resonance peaks. However, after treatment with $FeCl_3$, the $Fe^{3+}$ associated resonance was partially recovered. Moreover, the EDTA treatment of *Atrina* byssus compromised the mechanical properties of the byssal threads (Supplementary Table 1), suggesting the presence of metal–DOPA interactions. This result offers revealing insights into the $Fe^{3+}$–DOPA complexes in apfp-1 as it is homologous with the cuticle protein mfp-1, which plays an integral role of metal coordination chemistry in mechanical performance to create a stiff yet extensible coating for *Mytilus* byssus[22,23,30].

**Localization of apfp-1 mediates sugar–lectin interaction.** To localize apfp-1, a specific polyclonal antibody for apfp-1 was generated. Transmission electron microscopy (Supplementary Fig. 4) and fluorescence imaging of longitudinal and orthogonal cross-sections of the *Atrina* thread (Fig. 2f) revealed that apfp-1 is localized in the cuticle of the byssal threads (Supplementary Discussion). Given the cuticular localization of apfp-1 and colocalization of DOPA–$Fe^{3+}$ complexes (Fig. 2c,f), the similarities between mfp-1 and apfp-1 are impossible to ignore[17,18]. Nevertheless, the lectin domain distinguishes apfp-1 from mfp-1. As lectins are known to bind to sugars surrounding the cell membrane[31], we proceeded to identify the types of sugar specifically bound by apfp-1. Seven biotinylated lectins: (Concanavalin A (ConA): α-D-mannosyl and α-D-glucosyl residues branched α-mannosidic structures (high α-mannose type, or hybrid type and biantennary complex type N-Glycans), Biotinylated Dolichos Biflorus Agglutinin (DBA): α-linked N-acetylgalactosamine Biotinylated Soybean Agglutinin (SBA): α- or β-linked N-acetylgalactosamine, and to a lesser extent, galactose residues, Wheat Germ Agglutinin (WGA): GlcNAcβ1-4GlcNAcβ1-4GlcNAc, Neu5Ac (sialic acid), Ulex europaeus agglutinin (UEA1): Fucα1-2 Gal-R and Ricinus communis (RCA): Galβ1-4GalNAcβ1-R) were applied to sections performed through the adductor muscle and byssus. Only ConA labelling overlapped competitively with apfp-1 localization in *Atrina* tissue and byssus. Since the C-lectin binding domain is structurally predicted to be a mannose-binding lectin, apfp-1 is predicted to bind to mannose, the ligand for ConA. The confocal imaging shows apfp-1 bound to mannose in mucus, the living tissue or cell membranes (Fig. 2f). These findings suggest apfp-1 as an interfacial linker between the byssus and the living tissues via the asymmetric $Fe^{3+}$–DOPA and the sugar–lectin interactions. We speculated that the linkage to soft tissue is through the membrane bound sugars moieties (Fig. 2g). The $Fe^{3+}$–DOPA-mediated interactions between the byssus and mfp-1 were shown to be strong and reversible ($W_{ad} \sim -2.0$ mJ $m^{-2}$ to $-4.0$ mJ $m^{-2}$ using a surface forces apparatus (SFA))[30]. Lectin–sugar binding is also known to be strong and contributes to a variety of the cellular activities. Previous study using atomic force microscopy and SFA had shown specific and strong binding between lectin and its cognate sugar and a thorough understanding of this interaction holds the key to a better understanding of cell–cell interactions[32,33].

**Nano-mechanical quantification of the sugary interfaces.** To quantify how strong the sugar–lectin interaction is and how it

compares with the $Fe^{3+}$–DOPA complex, SFA was used to measure the adhesion between the sugar-binding domain of apfp-1 and mannose (Fig. 3a). SFA has been extensively used for measuring various types of biological interactions including specific and non-specific biological interactions and has the unique ability to provide a simultaneous direct measurement of the force $F$ as a function of the absolute surface separation $D$ measured *in situ* and in real time, with a force sensitivity of <10 nN and distance resolution of ~1 Å determined by multiple beam interferometry using FECO (Fringes of Equal Chromatic Order)[34].

Mannan is a branched form of mannose derived from yeast cell membranes and was used to measure adhesion between mannose and the lectin domain. There was no adhesion force between two opposing apfp-1 layers at pH 5.5 (Fig. 3b, yellow), however when mannan (average molecular weight 46,000 kDa) was injected (10 μM), the adhesion force increased to $W_{ad} \approx -0.7$ mJ m$^{-2}$ (Fig. 3b, cyan). Notably, when $Ca^{2+}$ cofactor (10 μM) was added, the measured adhesion energy increased to $W_{ad} \approx -1.7$ mJ m$^{-2}$ and was comparable with the $Fe^{3+}$–DOPA interaction in mfp-1 (ref. 30). Thus, addition of $Ca^{2+}$ cofactor enhanced adhesion (Fig. 3b, red). For a negative control, the adhesion between apfp-1 with $Ca^{2+}$ was measured without mannan, which resulted in $W_{ad} \approx -0.3$ mJ m$^{-2}$ (Fig. 3b, green). Interestingly, the adhesion force of lectin–mannan involving apfp-1, which has only one lectin domain, is comparable with the adhesion involving ~100 DOPA residues in mfp-1 protein or the DOPA-containing domain of apfp-1 (refs 29,30). Therefore, a considerable amount of force would be required to pull a lectin-covered thread through a saccharide covered tissue tunnel.

The finding that sugar–lectin binding is much stronger than $Fe^{3+}$–DOPA binding within the same byssus system is significant. As already mentioned, strong interfacial adhesion is required to prevent rubbing or sliding friction between the stiff cuticle of byssus and the soft living tissue of *Atrina*. Such friction would occur if the interfacial interactions between cuticle and soft tissue were too weak to prevent extensive slippage during load transfer and result in contact damage. apfp-1 is the ideal linker protein with bi-functionality, which are metal and sugar binding. The binding is weak enough to yield and reform, but not slip.

## Discussion

Combining the results of linker protein characterization, confocal imaging and SFA-based nano-mechanics, we have described an intriguing interaction between tissue and non-tissue components in *Atrina* in the absence of gradients. *Atrina* byssal threads are securely mounted within the soft tissue by two strong but yielding interfaces: the C-lectin binding domain is attached to mannose on the glycosylated cell membranes and serves as an interface between the adductor tissues and the embedded portion of byssus thread.

The robust interconnection of non-living tissue and soft tissue is mediated by lectin–sugar interactions. To the best of our knowledge, this study is the first to describe a bionic linker protein, apfp-1, that is involved in interconnecting non-living stiff biomaterial and soft tissue (Fig. 2g). Moreover, this work is also the first to explain the interconnection from the molecular point of view.

What are the implications of this finding? One of the great problems for interfaces between implants and living tissues is the friction between stiff and compliant surfaces that leads to contact damage[1,2,34]. The byssal thread–tissue interface is one in which a stiffness mismatch is mitigated by a large interfacial area that is populated by many sacrificial lectin–sugar bonds that dissipate energy but prevent sliding between surfaces. Specifically, *Atrina* has a two-phase mechanism in their load-bearing mitigation. When the load is transmitted directly to the tissue, *Atrina* exploits its high surface area associated with their embedded thread structure in the byssal groove for mitigation. The mitigation through reversible sacrificial lectin-type interaction would be used when the transmitted load is beyond its first phase handling capability. The advances in this area should inspire improved engineering of bionic implants and devices[35–37]. It is known that sugar-based molecules and polymers have been utilized as conventional adhesives since the beginning of the human history. Thus, a specific and strong adhesion between sugar and lectin in the living organism should be exploited as a key biomedical adhesive for the bionic interface. We envisage that this could shift the research paradigm from conventional research on sugar and lectin that have been focused on cell–cell communications and immune responses towards the focus on bionic implants and adhesives.

## Methods

**Fan shell mussels.** Live fan shell mussels (*A. pectinata*) were collected from a fan mussel farm in Gangjin, South Korea. Before biochemical and biomechanical analyses, the feet and silky byssus were carefully dissected from fresh fan shell mussels. The dissected feet were either used directly or stored at −80 °C until use. For mechanical and biochemical analyses, byssus threads were kept in artificial seawater at 4 °C until use.

**Nano-indentation of *Atrina* byssus and surrounding tissues.** Hardness value for the *Atrina* byssus was obtained by nano-indentation (Nano indenter G300, Aglient, Santa Clara, CA, USA) with a cube corner diamond tip. Nano-indentation was carried out on microtomed and surface polished surfaces of transverse cross-sections of the byssus and tissues embedded in Epofix. Test specimens hydrated in milliQ water for 24 h before the test. EPOXY used as a standard for the tip area

**a**

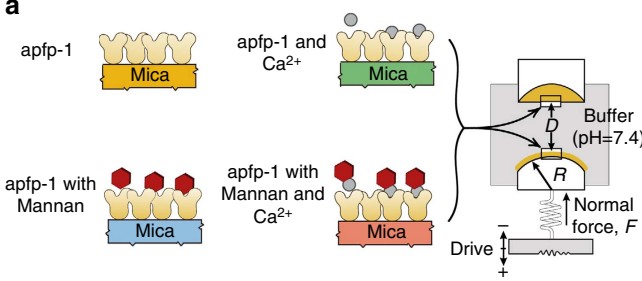

**b**

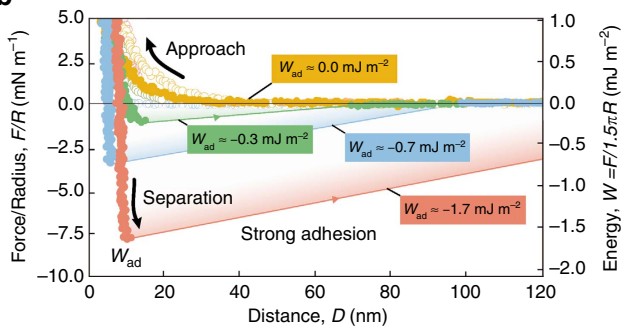

**Figure 3 | Biomechanics of apfp-1.** (**a**) Different mica surfaces coated to measure adhesion force between apfp-1 and apfp-1 (yellow), apfp-1 and apfp-1 in the presence of mannan (cyan), apfp-1 and apfp-1 in the presence of mannan and calcium (red), apfp-1 and apfp-1 in the presence of calcium (green). (**b**) Force-distance profiles from SFA measurements. The measured force, $F/R$ (normalized by the radius of curvature, $R$), is denoted in the ordinate at the left, whereas the corresponding interaction energy per unit area, $W$, between two flat surfaces, defined by $W = F/1.5\pi R$ is on the right. $F_{ad}$ and $W_{ad}$ are the adhesion force and the adhesion energy, respectively.

function. Indentation depth 2 μm and all load–displacement curves were analysed using the method described by Oliver and Pharr[38].

**Characterization of tensile properties of *Atrina* fibre.** The tensile properties of the *A. pectinata* byssus threads were measured under wet conditions using the universal tensile tester (UTS, Instron, USA). The proximal part of the threads was used for the analyses. In the case of wet conditions, threads were kept in stirring artificial seawater at room temperature with or without the additional incubation with 50 mM EDTA in 0.1 M sodium acetate buffer pH 5.5 for 24 h. Byssus threads were loaded to failure at a strain rate of $0.5\,mm\,s^{-1}$ with the use of a 10 N load cell. The humidity of the chamber was maintained at 99% by gently spraying a fine water mist from a humidifier during the wet condition tests. Strain energy density calculations were performed using SigmaPlot version 12.0 (Systat Software, Inc., San Jose, CA, USA).

**Biochemical analysis.** Byssus thread samples were incubated in 4 M urea/5% acetic acid solution for 48 h to remove contaminants, washed thoroughly with deionized water, and then freeze-dried, and their dry weight recorded. Dried threads were ground to powder with a mortar and pestle. Extraction of total lipids was performed in a mixture of methanol and chloroform, and the lipid content was determined by a previously reported method[39]. Total protein content was measured by the Lowry method[40] and by amino-acid analysis after sample preparation by acid hydrolysis[12]. Carbohydrates were extracted in 15% trichloroacetic acid from powdered thread samples and their content was determined by the phenol–sulfuric acid method[41]. Five samples were used for each component determination. For the Arnow assay of byssus and foot[42], the tissues were sequentially immersed for 5 min in each of the following solutions: 0.5 N hydrochloric acid, 1.45 M sodium nitrite/0.41 M sodium molybdate and 1 N sodium hydroxide.

**Inductively coupled plasma spectroscopy.** The byssus thread was dissected from the rest of the thread and rinsed three times with distilled water. Byssal threads from the three sampling dates were pooled for spawned and unspawned mussels to get the 3–4 mg of byssus needed for analysis of metal. Chemical element content of the byssal threads was determined using an IRIS advantage (Thermo, Franklin, MA, USA) ICP in normal mode and with a micro-nebulizer. Analyses were done after acid dissolution of byssal threads with 0.1 ml of concentrated nitric acid (70%) and 0.1 ml of hydrogen peroxide, placed in a water bath for 1 h at 60 °C. Elements detected are Fe, Ca and Zn. To correct for mass bias and instrument drift, a 2% $HNO_3$ blank solution was run periodically.

**Protein purification.** For protein extraction, frozen fan shell mussel feet were homogenized in a glass tissue grinder with 5% acetic acid containing two protease inhibitors ($1\,\mu g\,ml^{-1}$ leupeptin and pepstatin A) and 1 mM EDTA. The homogenate was centrifuged at 20,000*g*, 4 °C for 30 min. Perchloric acid was added drop-wise to the recovered supernatant to a final concentration of 1.4% (w/v). The mixture was further stirred for 10 min, and then centrifuged again at 20,000*g*, 4 °C for 30 min. The collected supernatant was dialyzed against 5% acetic acid overnight, then after addition of 2% acetonitrile, freeze-dried[43] (Supplementary Discussion).

**Gel permeation and HPLC.** Freeze-dried *Atrina* proteins were resuspended in 500 μl 5% acetic acid, centrifuged at 20,000*g*, 4 °C for 30 min to remove any precipitate, then injected to gel filtration column (Shodex KW 803) and eluted with 5% acetic acid at $0.5\,ml\,min^{-1}$ at room temperature and monitored at 280 nm. Collected GPC fractions were examined by AU-PAGE for the presence of DOPA-containing proteins[44]. Fractions containing the DOPA-containing proteins were further analysed in a C-8 reverse phase HPLC using an acetonitrile gradient in water with 0.1% trifluoroacetic acid at a flow rate of $1\,ml\,min^{-1}$. Separated proteins were freeze-fried and stored at −80 °C. A small aliquot (100 μl) from each collected peak was freeze-dried separately to be assayed by AU-PAGE for protein separation.

**Protein electrophoresis.** *Atrina* foot proteins were separated on polyacrylamide gels containing 5% acetic acid and 8 M urea (AU-PAGE), and then transferred to polyvinylidene difluoride (PVDF) membranes (Immobilon P, Millipore) by electrophoresis in 0.7% acetic acid, 200 mA, for 2 h. Proteins were stained with Coomassie Blue R-250 or Coomassie Blue G-250, the latter case when staining PVDF membrane. DOPA-containing proteins were stained with nitro blue tetrazolium redox cycling stain[45]. Protein bands of interest were excised and prepared for amino-acid analysis and peptide sequencing based on electrospray ionization-tandem mass spectrometry (ESI-MS/MS, Thermo, Franklin, MA, USA). *Atrina* foot proteins were also separated on SDS–PAGE, then transferred to PVDF membrane, and sent for N-terminal sequencing by Edman degradation.

**Protein and peptide sequencing.** N-terminal sequencing of excised PVDF membrane bands containing the protein of interest was done using automated

Edman degradation. Also, the excised PVDF membrane bands were transferred into hydrolysis vials containing 100 mM Tris ascorbate pH 7.5, 1 mM $CaCl_2$, $0.1\,mg\,ml^{-1}$ trypsin. The vials were then flushed with Argon and sealed by flame. Digestion was carried out for 12 h at room temperature. Digestion reaction was terminated by adding glacial acetic acid to a final concentration of 5%. Peptide sequencing was carried out in ESI-MS/MS (Thermo). To confirm the sequence, RT-PCR was run and a cDNA sequence of apfp-1 was obtained. After obtaining the deduced cDNA sequence of apfp-1, additional Edman sequencing of the purified protein was performed twice; the obtained Edman sequencing data was ASPVPPVD. The speculated reason for this discrepancy of the sequence between protein sequence translated from cDNA and Edman sequencing is highly due to the post-translational modification of tyrosine, which results in the difference of Y and P. As such, the following sequence, AS′X′VPPVD is used for the discrepancy part of the protein sequence (Supplementary Discussion).

**Mass spectrometry.** The masses of the two HPLC fractions analysed were determined by MALDI-TOF mass spectrometry (Applied Biosystems 4700, USA). The MALDI matrix was prepared by dissolving CHCA (a-cyano-4-hydroxycinnamic acid) ($7\,mg\,ml^{-1}$) in 0.1% trifluoroacetic acid (TFA)/50% acetonitrile (ACN). Purified proteins were dissolved in this matrix solution in a ratio of 10:1 matrix:sample. About 1 μl of sample was analysed (Supplementary Discussion).

**Raman spectroscopy.** A purified apfp-1 ($1\,mg\,ml^{-1}$) was added with a solution of $FeCl_3$ in 10 mM Bis-Tris (pH 5.5) and allowed to equilibrate for 10 min. The protein precipitated as the pH was raised to ∼8.0 with 0.1 M NaOH. A droplet was allowed to evaporate on a glass slide, and Raman spectra (30 accumulations with 1 s integration time each) were taken from the precipitated protein residue at the edges. Raman spectra were also measured from the protein solution before iron complexation and from solutions of apfp-1 in which $Fe^{3+}$ was not limiting for comparison (no EDTA added in the protein extraction). For Raman spectroscopy, a continuous laser beam was focused on the sample through a confocal Raman microscope (model CRM200, WITec, Ulm, Germany) equipped with a piezo scanner (model P-500, Physik Instrumente, Karlsruhe, Germany). The diode-pumped, 785-nm near-infrared laser excitation (Toptica Photonics AG, Graefelfing, Germany) was used in combination with a 100× oil immersed (Nikon, numerical aperture (NA) = 1.25) microscope objective. Laser power ranging between 15 and 30 milliwatts was used for all measurements. The spectra were acquired using an air-cooled charge-coupled device (DU401A-DR-DD, Andor, Belfast, North Ireland) behind a grating ($300\,g\,mm^{-1}$) spectrograph (Acton, Princeton Instruments, Trenton, NJ, USA) with a $6\,cm^{-1}$ spectral resolution. Software ScanCtrlSpectroscopyPlus (version 1.38, Witec) was used for measurement setup. Raman spectra were processed and analysed with Witec Project software (version 2.02). Raman spectra were background subtracted and lightly smoothed using the first order polynomial function and 9-point Savitzky-Golay filter (4th order polynomial), respectively.

**Lectin histochemistry.** *A. pectinata* adductor muscle, and byssal threads were fixed in 4% paraformaldehyde in sodium phosphate buffer (PBS solution, pH 7.4), rinsed in PBS solution, dehydrated through an ethanol series, embedded in paraffin, and cut into 5 μm-thick sections with a Microm HM 340 E microtome. The sections were subjected to an indirect lectin-labelling method according to the following protocol. Sections were blocked for 30 min in Tris-buffered saline, pH 8.0, containing 0.05% Tween 20 and 3% bovine serum albumin (TBS-T-BSA). The seven biotinylated lectins used for lectin staining (Concanavalin A (ConA): α-D-mannosyl and α-D-glucosyl residues branched α-mannosidic structures (high α-mannose type, or hybrid type and biantennary complex type N-Glycans), Biotinylated DBA: α-linked N-acetylgalactosamine, Biotinylated SBA: α- or β-linked N-acetylgalactosamine, and to a lesser extent, galactose residues, WGA: GlcNAcβ1-4GlcNAcβ1-4GlcNAc, Neu5Ac (sialic acid), Ulex europaeus agglutinin (UEA1): Fucα1-2 Gal-R and Ricinus communis (RCA): Galβ1-4GalNAcβ1-R; Vector Laboratories) were diluted at a concentration of $25\,\mu g\,ml^{-1}$ in TBS-T-BSA and applied on the sections for 2 h at room temperature. After three washes in TBS-T, the sections were incubated for 1 h in Texas-Red-conjugated streptavidin (Vector Laboratories) diluted 1:100 in TBS-T-BSA. Following three final washes in TBS-T, they were mounted with Vectashield (Vector Laboratories). Control reactions were performed by substituting the lectins with TBS-T-BSA. For ConA, an additional control reaction was also performed by using the lectin saturated with 0.4 M Methyl α-D-mannopyranoside. Sections were observed using a Zeiss Axioscope A1 microscope equipped with an AxioCam ICc 3 camera.

**Transmission electron microscopy.** *Atrina* foot samples were frozen using a high pressure freezer (Leica EMPACT2) at a pressure of 2,000–2,050 bar. Freeze substitution was performed in anhydrous acetone (containing 1% OsO4 and 0.1% uranyl acetate) using a Leica EM AFS2 (automatic freeze substitution). The samples were kept at −85 °C for 3 days, and then at −60, −20 and 0 °C for one day at each temperature, and then exposed to room temperature. Spurr's resin was used for infiltration and embedding. The embedded samples were cut to make ultrathin

sections at 70–90 nm using a Reichert Ultracut S or Leica EM UC6 (Leica, Vienna, Austria), and the sections were collected using 100 mesh copper grids.

**apfp-1 Immunolocalization.** A polyclonal anti-apfp-1 antibody was raised against the peptide VLFPKSHKWFWGYGKKQCKWFD (Ontores Biotechnology, China) which is part of the apfp-1 CLECT domain. The specificity of the antibody was established using apfp-1 protein purified from the foot tissue. Antibody labelling was done with apfp-1 antibody. Twenty nanometer gold nano secondary antibody from donkey (Abcam) was used to detect apfp-1 in ultrathin sections of the byssus.

**Electron ionization-tandem mass spectrometry.** A purified protein (apfp-1) was reduced with 5 mM dithiothreitol for 30 min, and then alkylated with 25 mM iodoacetamide for 30 min at 25 °C in the dark room. The protein was resolved in a digestion solution (6 M urea and 40 mM ammonium bicarbonate) and digested by trypsin (Sequencing-grade, Promega, Madison, WI, USA) for 12 h at 37 °C. The digested peptides were collected, desalted, separated with the C-18 column (Magic C18AQ 5 μ 200 Å particles, Michrom BioResources, Auburn, CA, USA), and analysed on a linear ion trap LTQ XL mass spectrometer (Thermo) interfaced with anelectrospray ion source. The peptides were separated using a linear gradient of ACN/water (2–40% ACN in 90 min), containing 0.1% formic acid, at a flow rate of 260 nl min$^{-1}$. A full MS scan range was from 300 to 2,000 $m/z$, and MS/MS spectra were collected from the five most intense precursor ions. The collected MS/MS spectra were searched against the *Atrina* protein sequence database using SEQUEST algorithm in the Proteome Discoverer software (Thermo) with the following parameters: semi-trypsin, a mass tolerance of 2.0 and 1.0 Da for precursor and fragment ions, respectively. All peptide identifications by the software were confirmed by manual interpretation of the raw data.

**Amino-acid analysis.** Protein of interest from excised PVDF membrane were hydrolysed in 6 N HCl with 5% water saturated phenol. The reactions were flushed with Argon and vials sealed by flame. Hydrolysis reaction was carried out for 1 h at 156 °C. Afterwards, the samples were flash evaporated at 60 °C under vacuum and washed till dry twice with 1 ml milliQ water, then twice with 1 ml methanol. The samples were then resuspended in 250–500 μl SYKAM sample dilution buffer. Amino-acid analysis was performed with a ninhydrin-based SYKAM System S4300 Amino Acid Analyzer (SYKAM, Germany).

**Molecular cloning.** Total RNA was extracted from dissected mussel feet regions that tested positive in the Arnow assay using an RNeasy Mini Kit (QIAGEN, Valencia, CA, USA). Briefly, dissected tissue previously stored in RNAlater solution was disrupted and homogenized in a glass tissue grinder with a Wise Stir homogenizer stirrer. Afterwards, total RNA was extracted according to the protocol specified by the kit. First-strand cDNA synthesis and 3′ and 5′ RACEs were performed by using SMARTer RACE cDNA amplification kit (Clontech, Palo Alto, CA, USA). Degenerate primers were designed based on the sequenced tryptic peptides.

**Force versus distance profiles with surface forces apparatus.** A surface forces apparatus (SFA) was used to measure the interaction forces of protein films in different aqueous solutions. Detailed setup of the SFA system and its working principles can be found elsewhere[46]. Briefly, two back-silvered thin mica surfaces with thickness ∼1 to 5 μm were glued onto cylindrical silica disks with a radius $R \sim 2$ cm. The two disks were mounted in the SFA chamber in a crossed-cylinder configuration, the interaction of which is equivalent to that of a sphere with radius $R$ approaching a flat surface when the separation distance $D \ll R$. Desired buffer solutions were injected between the two surfaces for further force measurement.

The interaction forces of apfp-1 versus apfp-1, in buffer solutions with of $Ca^{2+}$ (10 μM) or mannan (10 μM) or both $Ca^{2+}$ and mannan were measured in PBS buffer (pH 7.4). The film thickness and separation distance can be obtained *in situ* by using an optical technique called multiple beam interferometry using fringes of equal chromatic order (FECO). The adhesion force $F_{ad}$ measured can be correlated to the adhesion energy per unit area of two flat surfaces $W_{ad}$ by:

$$F_{ad} = 1.5 * \pi * R * W_{ad}$$

**Data availability.** Accession codes: *Atrina pectinata* foot protein-1 (apfp-1) datasets were deposited at the Protein database of National Center for Biotechnology Information under the accession number AIW04139.1. (http://www.ncbi.nlm.nih.gov/protein/701959535). The authors declare that all other data supporting the findings of this study are available within the article and its Supplementary Information files or available from the authors upon reasonable request.

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

## Acknowledgements

This work was supported by the Marine Biotechnology program (Marine BioMaterials Research Center) funded by the Ministry of Oceans and Fisheries, Korea (D11013214H480000110, D.S.H.). This research was also supported by the National Research Foundation of Korea Grant funded by the Ministry of Science, ICT and Future Planning (MSIP) (NRF-2014R1A2A2A01006724 and NRF-2011-0029960, D.S.H.), Global PhD Fellowship Program (NRF-2011-0008261), Institute for Basic Science (IBS-R013-G1) through the National Research Foundation of Korea, and a NSERC Discovery Grant and a NSERC RTI Grant (for a surface forces apparatus) from the Natural Sciences and Engineering Research Council of Canada (H. Zeng). P.F is Research Director of the Fund for Scientific Research of Belgium (F.R.S.-FNRS). We thank Prof. Admir Masic (MIT) for helping with resonance Raman spectroscopy.

## Author contributions

H.Y.Y. and M.I. prepared the samples and designed the experiments. J.H. conducted the SFA measurements, E.H. conducted the lectin imaging, S.K. performed the mechanical measurements, S.R. performed the ESI-MS/MS measurements, J.H.W. and D.S.H., conceived the project. M.F., P.F., H.Z., D.H., J.H.W. and D.S.H., advised the experimental design and analysed the results. All authors helped in the preparation of the manuscript.

## Additional information

**Competing financial interests:** The authors declare no competing financial interest.

