## [Peer review file · Nature Communications]

Reviewers' comments:

Reviewer #1 (Remarks to the Author):

The manuscript titled "Where stiff meets soft: sugary interfaces mitigate contact damage" by Hee Young Yoo et al described the unique structure of new byssal threads proteins from the fan shell *Atrina pectinata*, apfp-1, and their mechanical properties and localization in *Atrina* thread.

Although this article manuscript contains interesting findings, severe problems for publishing are exist.

Major points:

Although authors mentioned "ASPVPPVD as the starting sequence, which matches that of the cDNA deduced sequence after the signal peptide" in text (page 7, line 6), N-terminal amino acid sequence derived from cDNA-deduced protein (ASYVPPVD in Fig S3) was different from that from Edman sequencing, ASPVPPVD. This discrepancy was severer mistake because author discussed the modification of Tyr and Pro residues based on the mass differences between cDNA and ToF MS data (page 7, lines 8-16).

Furthermore, ToF MS data of purified apfp-1 showed two major mass peaks, 38705.4 and ca.34000, indicating that the purified apfp-1 sample is heterogeneity containing two major proteins (Fig. S1). It seems to me that SDS-PAGE (AU-PAGE) could not separate these proteins because apparent molecular mass of protein(s) band was 34 kDa.

Thus, other experiments using this sample such as biomechanics (SFA system), amino acid analysis for quantification of DOPA, and Raman spectroscopy have problems with non-negligible effect of impurity, 34 kDa protein.

To make clear these questions and suspicions, purified apfp-1 protein should be checked by 2D-PAGE and ToF MS after further purification steps (to remove 34 kDa protein).

Minor points:

1) Authors used mannan for SFA measurement at 10 μ M. Usually, molar concentration of mannan is difficult to determine because it is heterogeneous polymer of mannose.

Author should indicate how to calculate the concentration of mannan.

2) For lectin-histochemistry, authors used seven lectins, Con A, DBA, PNA, SBA, WGA, UEA I and RCA. However, these lectin indicated by only abbreviated name. Abbreviation should be defined, and their carbohydrate specificity for monosaccharide should be indicated. ConA: mannose, DBA: GalNAc, ...

3) Page 7, line1: 'molecular weight' should be 'molecular mass'.

4) Table S6: Glycine max is soybean, not mussels. So, the data from Glycine max (consensus PPVYK_, repeats (43)) in table S6 and corresponding reference, Ref. 18, should be deleted, or the title of Table S6 should be changed.

5) Figure S3: Author defined the repeating region from Val residue at the 32th position, however, N-terminal sequence Ala1 to Pro31 also contained repeating sequences, S/DYXPPV. Thus, N-terminal region (Ala1 to Pro31) should be included in repeating region.

6) Figure S4 shows immuno-gold staining in Figs, however, in the text, Fig 4 is data for CLECT (but disappeared). Which one is correct?

Reviewer #2 (Remarks to the Author):

This is a very interesting paper that reports the use of various experimental methods to support the conclusion that an interfacial *Atrina* byssal protein has domains that connect to the nonliving byssus through Fe-DOPA linkages and also to soft tissue through sugar based lectin-binding. In general, I believe this work is new and important and my comments are comparatively minor

The schematic in Figure 2g is a hypothesis with the linkage to soft tissue through membrane bound sugars moieties being the most speculative part of the hypothesis. I believe the authors' should note this speculation.

Better labeling of Figure 1 would allow the reader to understand the statement "Interestingly, the rooted thread portion originates in the byssal adductor muscle and proceeds from there along the byssal groove till it emerges from the living tissue and the shell (Fig. 1a,b)." Where is the living tissue? Where is the byssal groove? Is the byssal adductor muscle observable?

In the first section of the Results section, the authors make many statements that need appropriate citations. For instance, "Previous studies on byssal threads in the genus *Mytilus* invoked molecular gradients ..." (p. 4).

Supporting Information provided nice information on methods.

I am unfamiliar with the Raman spectral analysis used by the authors to detect Fe-catechol interactions (Figure 2c). Are there citations to justify the authors' peak assignments?

The numerical values of the adhesion energies reported in Figure 3b differ somewhat from those cited in the text?

The statement in the Discussion: "... the somewhat weaker C-lectin binding domain is attached to mannose on the glycosylated cell membranes..." seems to be unsupported by evidence that that the protein attaches to cell membrane and also seems to conflict with the author's earlier statement "that sugar-lectin binding is much stronger than Fe³⁺-DOPA binding".

Reviewer #3 (Remarks to the Author):

Overall, the paper is very interesting. The authors did quite a bit of characterization of the apfp-1 as a sacrificial interface between the stiff byssus and soft tissues to mitigate contact deformation. For the most part, manuscript is well written but there are some notable errors. There are several comments the authors should address prior to publication.

For introduction, p.3, the phrase "...have yet to be deciphered." sounds odd. In many cases the role of composition, organization of structural proteins etc., for these interfacial tissues are known. Do authors mean that these man-made materials have yet to replicate the properties of these tissues?

The introduction seems to end abruptly and failed to motivate how interfacial tissues relates to the results of this manuscript. The first paragraph of "Results" section appears to be the continuation of the introduction as opposed to be describing of results.

There are some grammar errors.

Reference 4 seems to be related to tendon insertion to bone. Which is not the same as tendon-muscle interface as stated. The insertion of tendon and ligament to bone utilizes entheses with also have gradation in composition of fibrocartilage, collagen fiber orientation, mineralization of soft tissues and insertion angle, etc. The second paragraph seems to be an oversimplified observation.

Second paragraph of p.4 appears to be missing references in the first couple of sentences.

Top of p.5, "from the gradients in Mytilus threads" what kind of "gradients"?

The authors suggest that the strong sacrificial interaction within the byssus channel between the byssus and the tissue of the channel aid in counter acting deformation damage. However, these tissues are also very soft (Table S1). Would there be a situation where instead of breaking the sacrificial bond and the load is directly transmitted to the tissue?

In Table S1, it is not clear which part of the organism was used to characterized the stiffness of the "soft tissue"

"Atomic Force Microscopy" the first letters probably do not need to be capitalized (p.9)

In figure 2f, what do the green boxes represent? They are labeled byssus but the figure caption do not specify how they were treated.

Addressing the specific points of review:

A. Summary of the key results

The summary is well written.

B. Originality and interest: if not novel, please give references

The manuscript is novel.

C. Data & methodology: validity of approach, quality of data, quality of presentation
Materials is very well characterized and the experiment was well designed.

D. Appropriate use of statistics and treatment of uncertainties

For Figure 2, it is not clear how many repeats were performed and if the reported work of adhesion is an average. Perhaps statistical analysis should be performed.

E. Conclusions: robustness, validity, reliability

Missing a conclusion.

F. Suggested improvements: experiments, data for possible revision

See above for the comments.

G. References: appropriate credit to previous work?

Yes

H. Clarity and context: lucidity of abstract/summary, appropriateness of abstract, introduction and conclusion

There were some minor issues with grammar, and the introduction could be more clear. Seems to missing conclusion.

Authors' comments

Reviewer #1 (Remarks to the Author):

The manuscript titled "Where stiff meets soft: sugary interfaces mitigate contact damage " by Hee Young Yoo et al described the unique structure of new byssal threads proteins from the fan shell *Atrina pectinata*, apfp-1, and their mechanical properties and localization in *Atrina* thread.

Although this article manuscript contains interesting findings, severe problems for publishing are exist.

Ans) We appreciate the constructive comments and suggestions of the reviewer #1. We agree with most of the comments made by the reviewer, most of which were asking for further clarification and more experimental details.

Major points:

1) Although authors mentioned "ASPVPPVD as the starting sequence, which matches that of the cDNA deduced sequence after the signal peptide" in text (page 7, line 6), N-terminal amino acid sequence derived from cDNA-deduced protein (ASYVPPVD in Fig S3) was different from that from Edman sequencing, ASPVPPVD. This discrepancy was severer mistake because author discussed the modification of Tyr and Pro residues based on the mass differences between cDNA and Tof MS data (page 7, lines 8-16).

Ans) Thank you for pointing this out. We appreciate the reviewer's concern on this. Because the reviewer has raised this point, we have run the RT-PCR again and confirmed that the cDNA sequence for the mismatching sequence is TAT, which translates into Y. Thus, from this result, we can confirm that the obtained cDNA sequence is correct.

Next, we again performed the Edman sequencing twice with the purified apfp-1 protein. In both batches, the eluted amino acid is in position around 'P' instead of position 'Y'. From this result, we deduced that there is a high possibility that this 'Y' is a modified tyrosine, which results in the elution time near to proline. Additionally, from the Edman sequencing we can confirm that this modified tyrosine is not DOPA by injecting DOPA standard.

We hope for the reviewer's understanding on this matter that until this modified tyrosine is accurately characterized in the future, we think it is more appropriate to write our peptide sequence as AS'X'PPVD and we have updated this accordingly throughout the manuscript. Further analysis to characterize this modified tyrosine is currently beyond the scope of this manuscript.

To further explain, despite having one sequence difference, we think that this would not result in any major issue based on the reasons given below.

(1) We had additionally performed amino acid analysis of purified apfp-1 (Table S3) whenever we had purified apfp-1. Also we confirmed that the amino acid composition of the purified apfp-1 and that of cDNA deduced sequence are roughly the same. We would like to note that the Trp cannot be analyzed in HCl based amino acid analysis used in this study.

(2) MADLI TOF mass data of the purified protein (Figure S1) has been constantly ~38 kDa in every batch that we had analyzed after purification.

(3) We had also confirmed the purity of the protein with SDS-PAGE and NBT-staining (Figure 1e).

(4) The apfp-1 was recognized with anti-apfp-1 (Figure 2f, Figure S4).

We do acknowledge that this point of mismatch in the sequence should have been clarified and discussed in the main text to avoid any confusion. We have included the following sentences on page 4 in the supporting information as below.

"To confirm the sequence, RT-PCR was run and a cDNA sequence of apfp-1 was obtained. After obtaining the deduced cDNA sequence of apfp-1, additional Edman sequencing of the purified protein was performed twice; the obtained Edman sequencing data was ASPVPPVD. The speculated reason for this discrepancy of the sequence between protein sequence translated from cDNA and Edman sequencing is highly due to the post-translational modification of tyrosine, which results in the difference of Y and P. As such, the following sequence, AS'X'VPPVD is used for the discrepancy part of the protein sequence."

And changed the sentence in page 7 in the main text as below.

"In addition, the 38 kDa protein from the major peak was subjected to Edman N terminal sequencing and revealed ASXVPPVD (where X denotes an unknown amino acid residue)

as the starting sequence, which matched that of the cDNA deduced sequence after the signal peptide.”

2) Furthermore, Tof MS data of purified apfp-1 showed two major mass peaks, 38705.4 and ca.34000, indicating that the purified apfp-1 sample is heterogeneity containing two major proteins (Fig. S1). It seems to me that SDS-PAGE (AU-PAGE) could not separate these proteins because apparent molecular mass of protein(s) band was 34 kDa.

Thus, other experiments using this sample such as biomechanics (SFA system), amino acid analysis for quantification of DOPA, and Raman spectroscopy have problems with non-negligible effect of impurity, 34 kDa protein.

To make clear these questions and suspicions, purified apfp-1 protein should be checked by 2D-PAGE and Tof MS after further purification steps (to remove 34 kDa protein).

Ans) Thank you for pointing this out. We truly apologize for this confusion that led to this comment. As the reviewer has rightly pointed out, the MALDI data in Fig. S1 shows two peaks at 38kDa and 34kDa. It is purely our oversight that the wrong figure – the mass spectrum of the partially purified apfp-1 was included in Fig. S1. We would like to clarify that our experiments for apfp-1 are all carried out using the clean and purified protein with a single peak at 38kDa. We have confirmed the purity of this protein through (1) amino acid analysis, (2) SDS-PAGE (3) NBT-staining, and (4) MALDI-TOF after purification of apfp-1. Therefore, we are confident that the protein we used is thoroughly purified. We apologize for this confusion and we have updated the correct MALDI-TOF data that shows only a single peak at 38kDa in Figure S1.

Figure S1. MALDI-TOF mass spectrometry of apfp-1 purified by HPLC.

Minor points:

1) Authors used mannan for SFA measurement at 10 μ M. Usually, molar concentration of mannan is difficult to determine because it is a heterogeneous polymer of mannose.

Author should indicate how to calculate the concentration of mannan.

Ans) Thank you for this comment. We agree with the reviewer that it is usually difficult to measure the molar concentration of mannan. Since the structure of mannan is branched and depending on the branching, the molar concentration would be different, thus it is indeed difficult to measure the exact concentration of the mannan. Because of this, we used the average molecular weight to measure 10 μ M for the SFA experiment. The average molecular weight of the mannan that we used was 46 000 kDa, which we

added 4.6mg in 10ml buffer for further SFA experiment. We have included this detail in our revised manuscript on Page 10 as below.

"There was no adhesion force between two opposing apfp-1 layers at pH 5.5 (Fig. 3b, yellow), however when mannan (average molecular weight 46 000 kDa) was injected (10 μ M), the adhesion force increased to $W_{ad} \approx 0.8$ mJ/m² (Fig. 3b, cyan)."

2) For lectin-histochemistry, authors used seven lectins, Con A, DBA, PNA, SBA, WGA, UEA I and RCA. However, these lectin indicated by only abbreviated name. Abbreviation should be defined, and their carbohydrate specificity for monosaccharide should be indicated. ConA: mannose, DBA: GalNAc, ...

Ans: Thank you for pointing this out and we fully agree that abbreviation of each lectin and its carbohydrate specificity should be indicated. For the reviewer's information, the full name of these seven lectins and their carbohydrate specificity for monosaccharide are: Concanavalin A (ConA): α -D-mannosyl and α -D-glucosyl residues branched α -mannosidic structures (high α -mannose type, or hybrid type and biantennary complex type N-Glycans), Biotinylated Dolichos Biflorus Agglutinin (DBA): α -linked N-acetylgalactosamine Biotinylated Soybean Agglutinin (SBA): α - or β -linked N-acetylgalactosamine, and to a lesser extent, galactose residues, Wheat Germ Agglutinin (WGA): GlcNAc β 1-4GlcNAc β 1-4GlcNAc, Neu5Ac (sialic acid), Ulex europaeus agglutinin (UEA1): Fuca1-2Gal-R and Ricinus communis (RCA): Gal β 1-4GalNAc β 1-R. We have added this full name of the lectins in the revised manuscript on Page 9 and page 5-6 of the supporting information as below.

"Seven biotinylated lectins: (Concanavalin A (ConA): α -D-mannosyl and α -D-glucosyl residues branched α -mannosidic structures (high α -mannose type, or hybrid type and biantennary complex type N-Glycans), Biotinylated Dolichos Biflorus Agglutinin (DBA): α -linked N-acetylgalactosamine Biotinylated Soybean Agglutinin (SBA): α - or β -linked N-acetylgalactosamine, and to a lesser extent, galactose residues, Wheat Germ Agglutinin (WGA): GlcNAc β 1-4GlcNAc β 1-4GlcNAc, Neu5Ac (sialic acid), Ulex europaeus agglutinin (UEA1): Fuca1-2Gal-R and Ricinus communis (RCA): Gal β 1-4GalNAc β 1-R) were applied to sections performed through the adductor muscle and byssus."

3) Page 7, line1: 'molecular weight' should be 'molecular mass'.

Ans) We have made the changes accordingly in the revised manuscript on Page 7 Line 1.

"The predicted molecular mass and theoretical pI of the cDNA deduced amino acid sequence without the signal peptide were 38485.8 Da and 9.47, respectively."

and the legend of Figure 1.

"(e) Gel electrophoresis purification of apfp-1. M: Molecular mass markers, 1: fractions corresponding to GPC major peak; 2: HPLC major peak obtained after injection of the GPC main peak, 3 Catechol-specific staining of apfp-1"

4) Table S6: Glycine max is soybean, not mussels. So, the data from Glycine max (consensus PPVYK_, repeats (43)) in table S6 and corresponding reference, Ref. 18, should be deleted, or the title of Table S6 should be changed.

Ans) We agreed with the comments and we had deleted glycine max from Table S6, as suggested.

Table S6. Consensus repeat sequences in fp1 proteins from several mussels.

Species	Consensus	Repeats	Reference
Atrina pectinate	VVPDYKP	(7)	Present
Atrina pectinate	VPKYK_	(4)	Present
Atrina pectinate	PVYK_	(6)	Present
Perna canaliculus	PYVK_	(72)	Zhao, 2005 ¹³
Aulacomya ater	AGYGGVK_		Burzio, 2000 ¹⁴
Trichomya hirsute	SYYPK_		Rzepecki, 1991 ¹⁵
Modiolus modiolus	SSYYPK_		Rzepecki, 1991 ¹⁵
Choromytilus choros	AKPSYPTGYKPPVK_		Burzio. 2000 ¹⁴
Mytilus edulis	AKPSYPPTYK_____	(71)	Filpula, 1990 ¹⁶
Mytilus galloprovincialis	AKPSYPPTYK_____	(85)	Inoue, 1994 ¹⁷

5) Figure S3: Author defined the repeating region from Val residue at the 32th position, however, N-terminal sequence Ala1 to Pro31 also contained repeating sequences, S/DYXPPV. Thus, N-terminal region (Ala1 to Pro31) should be included in repeating region.

Ans) Thank you for pointing out this repeating region. We have included the suggested repeating region in Figure 1f, 2a and Figure S3. Also, the caption for Figure S3 had been updated accordingly.

f

ESI-MS/MS: **ODYKOVPK** (O denotes hydroxyproline)
 Design primer from PDYKVPVK -> obtain cDNA deduced sequence
 cDNA sequence : 1056 bp Protein : 352 a.a, 38 kDa Name: **apfp-1**

Figure 1f

Figure 2a

Figure S3. The nucleotide sequence, amino acid sequences, and regions in apfp-1. **(a)** Nucleotide and amino acid sequences of cloned apfp-1. Signal peptide is highlighted in blue, Glycine-rich region is highlighted in yellow, CLECT domain is highlighted in purple and the peptides isolated following ESI-MS/MS are underlined in red. The nucleotide sequence for apfp-1 gene has been deposited in the GenBank database under GenBank Accession Number KF951620. **(b)** The apfp-1 sequence between signal peptide and glycine rich region is the repeating region, which is highlighted in green. The grey rectangle represents VVPDYKP repeat, the sky blue rectangle represents the VPKYK repeat, the blue rectangle represents the P_YK repeat and the red rectangle represents the S/DY_PPV repeat.

6) Figure S4 shows immuno-gold staining in Figs, however, in the text, Fig 4 is data for CLECT (but disappeared). Which one is correct?

Ans) The correct label showing CLECT is Figure S3. We have updated the text accordingly on page 11 in the supporting information.

"The C-terminus of the protein is predicted by the conserved domain search on NCBI to consist of a Ca-dependent or C-type lectin module (CLECT domain) (Fig. S3A)."

Reviewer #2 (Remarks to the Author):

This is a very interesting paper that reports the use of various experimental methods to support the conclusion that an interfacial *Atrina* byssal protein has domains that connect to the nonliving byssus through Fe-DOPA linkages and also to soft tissue through sugar based lectin-binding. In general, I believe this work is new and important and my comments are comparatively minor

1) The schematic in Figure 2g is a hypothesis with the linkage to soft tissue through membrane bound sugars moieties being the most speculative part of the hypothesis. I believe the authors' should note this speculation.

Ans) We thank you for this comment. As the reviewer has mentioned, the schematic in Fig. 2g is indeed our hypothesis. We have added the following sentence on Page 9 of the main text.

"We speculated that the linkage to soft tissue is through the membrane bound sugars moieties (Fig. 2g)."

2) Better labeling of Figure 1 would allow the reader to understand the statement "Interestingly, the rooted thread portion originates in the byssal adductor muscle and proceeds from there along the byssal groove till it emerges from the living tissue and the shell (Fig. 1a,b)." Where is the living tissue? Where is the byssal groove? Is the byssal adductor muscle observable?

Ans) Thank you for pointing this out. Based on reviewer's suggestion, we have revised Figure 1a and b, which we think can clearly indicate the location of the living tissue, groove, and the adductor mussel.

Figure 1a, b

3) In the first section of the Results section, the authors make many statements that need appropriate citations. For instance, "Previous studies on byssal threads in the genus *Mytilus* invoked molecular gradients ..." (p. 4).

Ans) Thank you for pointing this out. We have included the appropriate citation.

Atrina uses a byssus to anchor itself to available hard objects in the benthic sediment [1,2]

Water flow over the exposed shell results in lift and drag forces that tug on the attached threads. The threads transfer and dissipate the loads to the interconnecting tissue without incurring any damage [2-5].

Previous studies on byssal threads in the genus *Mytilus* invoked molecular gradients along the axis of each thread to produce a stiffness gradient to moderate the stiffness mismatch between the threads and where these meet in the stem before entering the living tissue [3].

Specifically, two hybrid collagens in each thread, preCOL-D (collagen + silk domains) and preCOL-P (collagen + elastin domains), are self-assembled in such a way that the stiffer preCOL-D predominates distally, whereas the more compliant preCOL-P prevails approaching the living tissue [3-5].

Reference

[1] Pearce, Trevor, and Michael LaBarbera. "Biomechanics of byssal threads outside the Mytilidae: *Atrina rigida* and *Ctenoides mitis*." *Journal of Experimental Biology* 212.10 (2009): 1449-1454.

[2] Mascolo, J. M., and J. H. Waite. "Protein gradients in byssal threads of some marine bivalve molluscs." *Journal of Experimental Zoology* 240.1 (1986): 1-7.

[3] Qin, Xiao-Xia, and J. Herbert Waite. "A potential mediator of collagenous block copolymer gradients in mussel byssal threads." *Proceedings of the National Academy of Sciences* 95.18 (1998): 10517-10522.

[4] Harrington, Matthew J., and J. Herbert Waite. "Holdfast heroics: comparing the molecular and mechanical properties of *Mytilus californianus* byssal threads." *Journal of Experimental Biology* 210.24 (2007): 4307-4318.

[5] Coyne, Kathryn J., Xiao-Xia Qin, and J. Herbert Waite. "Extensible collagen in mussel byssus: a natural block copolymer." *Science* 277.5333 (1997): 1830-1832.

4) Supporting Information provided nice information on methods. I am unfamiliar with the Raman spectral analysis used by the authors to detect Fe-catechol interactions (Figure 2c). Are there citations to justify the authors' peak assignments?

Ans) Thank you for your comment regarding this point. We acknowledge that proper citation from previous literature on the Raman spectral analysis used in this paper should be given. We have added this reference which justify the peak assignment we found in this work. This reference has been added on Page 8 to our manuscript.

The formation of Fe³⁺-DOPA complexes was detected by resonance Raman spectroscopy [1,2] of purified apfp-1 protein (Fig. 2b,c).

Reference

[1] Harrington, M. J., Masic, A., Holten-Andersen, N., Waite, J. H., & Fratzl, P. (2010). Iron-clad fibers: a metal-based biological strategy for hard flexible coatings. *Science*, 328(5975), 216-220,

[2] Hwang, D. S., Zeng, H., Masic, A., Harrington, M. J., Israelachvili, J. N., & Waite, J. H. (2010). Protein-and metal-dependent interactions of a prominent protein in mussel adhesive plaques. *Journal of biological chemistry*, 285(33), 25850-25858.

5) The numerical values of the adhesion energies reported in Figure 3b differ somewhat from those cited in the text?

Ans) We have corrected our mistakes accordingly in the main text on Page 10.

"There was no adhesion force between two opposing apfp-1 layers at pH 5.5 (Fig. 3b, yellow), however when mannan (average molecular weight 46 000 kDa) was injected (10 μM), the adhesion force increased to $W_{ad} \approx -0.7 \text{ mJ/m}^2$ (Fig. 3b, cyan). Notably, when Ca²⁺ cofactor (10 μM) was added, the measured adhesion energy increased to $W_{ad} \approx -1.7 \text{ mJ/m}^2$ and was comparable with the Fe³⁺-DOPA interaction in mfp-1. Thus, addition of Ca²⁺ cofactor enhanced adhesion (Fig. 3b, red). For a negative control, the adhesion between apfp-1 with Ca²⁺ was measured without mannan, which resulted in $W_{ad} \approx -0.3 \text{ mJ/m}^2$ (Fig. 3b, green)."

6)The statement in the Discussion: "... the somewhat weaker C-lectin binding domain is attached to mannose on the glycosylated cell membranes..." seems to be unsupported by evidence that that the protein attaches to cell membrane and also seems to conflict with the author's earlier statement "that sugar-lectin binding is much stronger than Fe³⁺-DOPA binding".

Ans) Thank you for your comment. We have revised the sentence of Page 11.

"*Atrina* byssal threads are securely mounted within the soft tissue by two strong but yielding interfaces: the C-lectin binding domain is attached to mannose on the glycosylated cell membranes and serves as an interface between the adductor tissues and the embedded portion of byssus thread."

Reviewer #3 (Remarks to the Author):

Overall, the paper is very interesting. The authors did quite a bit of characterization of the apfp-1 as a sacrificial interface between the stiff byssus and soft tissues to mitigate contact deformation. For the most part, manuscript is well written but there are some notable errors. There are several comments the authors should address prior to publication.

1) For introduction, p.3, the phrase "...have yet to be deciphered." sounds odd. In many cases the role of composition, organization of structural proteins etc., for these interfacial tissues are known. Do authors mean that these man-made materials have yet to replicate the properties of these tissues?

Ans) Thank you for your comment, we have changed the sentence as below on Page 3.

"These natural interfaces are robust, dynamic, and durable - all desirable properties for bionic devices- but on their transferability to engineer bionic devices have yet to be fully understood."

2) The introduction seems to end abruptly and failed to motivate how interfacial tissues relates to the results of this manuscript. The first paragraph of "Results" section appears to be the continuation of the introduction as opposed to be describing of results.

Ans) Thank you for this comment. We acknowledge that the introduction does seem to end appropriately. The first paragraph of the "Results: Mechanical mismatch between living and non-living material" should be part of the introduction. And the main result for the Mechanical mismatch between living and non-living material should begin with the sentence, "Previous studies on byssal threads..." We have updated the manuscript accordingly.

There are some grammar errors.

Ans) Thank you for pointing this out. We have engaged professional English editing

service to address this. Due to the many changes after the editing, we are unable to specify each of the corrected grammar here.

3) Reference 4 seems to be related to tendon insertion to bone. Which is not the same as tendon-muscle interface as stated. The insertion of tendon and ligament to bone utilizes enthesis with also have gradation in composition of fibrocartilage, collagen fiber orientation, mineralization of soft tissues and insertion angle, etc. The second paragraph seems to be an oversimplified observation.

Ans) Thank you for pointing out the mismatch between the text and the reference. Reference 4 is related to tendon-bone interface and Reference 5 is related to fibrocartilages, thus muscle-cartilages interface. So we have changed the manuscript on Page 3 as following:

"There is growing belief that the best engineered interfaces are those inspired by nature (bionic), such as between bone and cartilage ^{4,5}, nail and skin ⁶, and more recently, squid beak and muscle⁷. "

To:

"There is growing belief that the best engineered interfaces are those inspired by nature (bionic), such as between bone, cartilage and muscle ^{4,5}, nail and skin ⁶, and more recently, squid beak and muscle ⁷. "

so this will cover both types of interfaces.

4) Second paragraph of p.4 appears to be missing references in the first couple of sentences.

Ans) Thank you for this comment. This in fact is also pointed out by Reviewer 2 and we have added the appropriate references.

Atrina uses a byssus to anchor itself to available hard objects in the benthic sediment [1,2]

Water flow over the exposed shell results in lift and drag forces that tug on the attached threads. The threads transfer and dissipate the loads to the interconnecting tissue without incurring any damage [2-5].

Previous studies on byssal threads in the genus *Mytilus* invoked molecular gradients along the axis of each thread to produce a stiffness gradient to moderate the stiffness mismatch between the threads and where these meet in the stem before entering the living tissue [3].

Specifically, two hybrid collagens in each thread, preCOL-D (collagen + silk domains) and preCOL-P (collagen + elastin domains), are self-assembled in such a way that the stiffer preCOL-D predominates distally, whereas the more compliant preCOL-P prevails approaching the living tissue [3-5].

Reference

[1] Pearce, Trevor, and Michael LaBarbera. "Biomechanics of byssal threads outside the Mytilidae: *Atrina rigida* and *Ctenoides mitis*." *Journal of Experimental Biology* 212.10 (2009): 1449-1454.

[2] Mascolo, J. M., and J. H. Waite. "Protein gradients in byssal threads of some marine bivalve molluscs." *Journal of Experimental Zoology* 240.1 (1986): 1-7.

[3] Qin, Xiao-Xia, and J. Herbert Waite. "A potential mediator of collagenous block copolymer gradients in mussel byssal threads." *Proceedings of the National Academy of Sciences* 95.18 (1998): 10517-10522.

[4] Harrington, Matthew J., and J. Herbert Waite. "Holdfast heroics: comparing the molecular and mechanical properties of *Mytilus californianus* byssal threads." *Journal of Experimental Biology* 210.24 (2007): 4307-4318.

[5] Coyne, Kathryn J., Xiao-Xia Qin, and J. Herbert Waite. "Extensible collagen in mussel byssus: a natural block copolymer." *Science* 277.5333 (1997): 1830-1832.

5) Top of p.5, "from the gradients in *Mytilus* threads" what kind of "gradients"?

Ans) Thank you for pointing this out. The gradient we are referring to is the molecular and mechanical gradients [1],[2],[3],[4] due to the molecular properties of the three collagenous proteins and the distribution of metal ions in the byssus. We have added "the molecular and mechanical gradients [1],[2],[3],[4]" in the main manuscript Page 5.

Reference

[1] Sun, Cheng Jun, and J. Herbert Waite. "Mapping chemical gradients within and along a fibrous structural tissue, mussel byssal threads." *Journal of Biological Chemistry* 280.47 (2005): 39332-39336.

[2] Coyne, Kathryn J., Xiao-Xia Qin, and J. Herbert Waite. "Extensible collagen in mussel byssus: a natural block copolymer." *Science* 277.5333 (1997): 1830-1832.

[3] Qin, Xiao-Xia, and J. Herbert Waite. "A potential mediator of collagenous block copolymer gradients in mussel byssal threads." *Proceedings of the National Academy of Sciences* 95.18 (1998): 10517-10522.

[4] Harrington, Matthew J., and J. Herbert Waite. "Holdfast heroics: comparing the molecular and mechanical properties of *Mytilus californianus* byssal threads." *Journal of Experimental Biology* 210.24 (2007): 4307-4318.

6) The authors suggest that the strong sacrificial interaction within the byssus channel between the byssus and the tissue of the channel aid in counter acting deformation damage. However, these tissues are also very soft (Table S1). Would there be a situation where instead of breaking the sacrificial bond and the load is directly transmitted to the tissue?

Ans) Thank you for pointing out this. Due to the fact that *Atrina* threads exploit the high surface area associated with the embedding of nearly 10 cm of each thread in the byssal

groove, in the first phase, the load is transmitted directly to the tissue. But if the load is greater than what can be transferred to the tissue, this is when the reversible sacrificial lectin-type interactions comes into action to mitigate the potential damage incurred to the bond when the load is beyond their handling capability after the first phase. We have added the description of this on Page 12 of the main text.

"Specifically, *Atrina* has a two-phase mechanism in their load bearing mitigation. When the load is transmitted directly to the tissue, *Atrina* exploits its high surface area associated with their embedded thread structure in the byssal groove for mitigation. The mitigation through reversible sacrificial lectin-type interaction would be used when the transmitted load is beyond its first phase handling capability."

7) In Table S1, it is not clear which part of the organism was used to characterized the stiffness of the "soft tissue"

Ans) The part that we are referring to is the adductor muscle of *Atrina*. We have revised this point accordingly. We have revised the caption of Table S1 accordingly.

8) "Atomic Force Microscopy" the first letters probably do not need to be capitalized (p.9)

Ans) We have changed it to small letters on Page 10 as below.

"Previous study using atomic force microscopy (AFM) and SFA had shown specific and strong binding between lectin and its cognate sugar and a thorough understanding of this interaction holds the key to a better understanding of cell-cell interactions."

9) In figure 2f, what do the green boxes represent? They are labeled byssus but the figure caption do not specify how they were treated.

Ans) Thank you for pointing this out. The green box represents the auto-fluorescence of the byssus which is the location that shows where the byssus fiber is. We have updated the caption for figure 2f.

"Green box represents the auto-fluorescence of the byssus which is the location that shows where the byssus fiber is. Red box represents the apfp-1 antibody to detect apfp-1, and with ConA to detect mannose."

Reviewer #1 (Remarks to the Author)

The revised manuscript has been corrected with sufficient condition. Thus, this manuscript is now acceptable for Nature Communications